# Health-Literate Healthcare Organizations and Quality of Care in Hospitals: A Cross-Sectional Study Conducted in Tuscany

**DOI:** 10.3390/ijerph17072508

**Published:** 2020-04-06

**Authors:** Guglielmo Bonaccorsi, Anna Romiti, Francesca Ierardi, Maddalena Innocenti, Marco Del Riccio, Silvia Frandi, Letizia Bachini, Patrizio Zanobini, Fabrizio Gemmi, Chiara Lorini

**Affiliations:** 1Department of Health Science, University of Florence, 50134 Florence, Italy; guglielmo.bonaccorsi@unifi.it (G.B.); maddalena.innocenti@unifi.it (M.I.) silvia.frandi@gmail.com (S.F.); patrizio.zanobini@unifi.it (P.Z.); chiara.lorini@unifi.it (C.L.); 2Department of Experimental and Clinical Medicine, University of Florence, 50134 Florence, Italy; anna.romiti@unifi.it; 3Quality and Equity Unit, Regional Health Agency of Tuscany, 50141 Florence, Italy; francesca.ierardi@ars.toscana.it (F.I.); letizia.bachini@ars.toscana.it (L.B.); fabrizio.gemmi@ars.toscana.it (F.G.)

**Keywords:** HLHO, perceived quality of care, health literacy

## Abstract

The concept of Health-Literate Healthcare Organization (HLHO) concerns the strategies by which healthcare organizations make it easier for people to navigate, understand, and use information and services to take care of their health. The aims of this study were to validate the HLHO-10 questionnaire in the Italian language; to measure the degree of implementation of the 10 attributes of HLHOs in a sample of hospitals placed in Tuscany; and to assess the association between the degree of implementation of the 10 attributes of HLHOs and the perceived quality of care. This was a cross-sectional study where data were collected using a self-administered questionnaire including three sections: a descriptive section, a section focused on the perceived quality, and the Italian version of the HLHO-10 questionnaire. A total amount of 405 healthcare managers answered the questionnaire (54.9%). The analysis shows that the HLHO score is significantly associated with the type of hospitals: accredited private hospitals have higher HLHO scores. Moreover, the perceived quality increases with the increasing of the HLHO score, with the highest coefficient for local public hospitals. In conclusion, Organizational Health Literacy culture should be an integral element for the management to improve the quality of care.

## 1. Introduction 

Health literacy (HL) is a multidimensional concept which was originally developed in the 1970s [1] and has been moving from an individual to a public health perspective [2]. In one of its definitions, HL is described as the degree to which individuals can obtain, process, and understand the basic health information and services they need to make appropriate health decisions [3]. It involves the interaction with healthcare services and other societal institutions [4], so it does not only concern individuals, but it is also embedded in organizational structures and societal systems [5]. HL is increasingly recognized as a characteristic related to families, communities, and organizations providing health and social services, instead of just an individual trait [6]. For this reason, attention has also shifted to the specific context in which care is provided: patients’ ability to understand medical information and navigate the care-seeking process is related to the demands that health delivery systems place on them, and the specific organizational context where care is provided may contribute to compensating for patients’ limited HL [7,8,9]. In fact, HL is nowadays considered the product of the interaction between individuals’ capacities and the health literacy-related demands and complexities of the healthcare system [10].

The importance of HL has not only been recognized in terms of health outcomes, but also in terms of economic results [11]. The instruments to increase HL levels are considered “inexpensive and easy to implement”, but at the same time, capable of reducing costs and improving outcomes in terms of disease prevention [12,13] by means of increased adherence to potentially life-saving screening [14], decrease in hospitalization rates [15,16], and appropriate use of healthcare services [17].

To assess how healthcare organizations perform in dealing with patients’ HL issues, the concept of health-literate healthcare organizations (HLHOs)—or organizational HL (OHL)—has been proposed by Brach [18]. This concept concerns the strategies by which healthcare organizations make it easier for people, in spite of their level of HL, to navigate, understand, and use information and services to take care of their health, providing clear services and information for all those seeking services to find and understand, to assist them in the decisions they make, and to eliminate existing barriers in these issues [19].

Specifically, the 10 attributes of an HLHO have been described as follows [18]:Has leadership that makes health literacy integral to its mission, structure, and operations.Integrates health literacy into planning, evaluation measures, patient safety, and quality improvement.Prepares the workforce to be health literate and monitors progress.Includes populations served in the design, implementation, and evaluation of health information and services.Meets the needs of populations with a range of health literacy skills while avoiding stigmatization.Uses health literacy strategies in interpersonal communications and confirms understanding at all points of contact.Provides easy access to health information and services and navigation assistance.Designs and distributes print, audiovisual, and social media content that is easy to understand and act on.Addresses health literacy in high-risk situations, including care transitions and communications about medicines.Communicates clearly what health plans cover and what individuals will have to pay for services.

Therefore, an HLHO is designed to help build a person-centered, evidence-based, and quality-driven healthcare system [20]. Despite this, current healthcare systems are nowadays designed on the assumption that their users have adequate HL levels [21].

To measure the degree of implementation of these attributes in healthcare organizations, Kowalski et al. [22] developed a measurement tool—the 10-item questionnaire (HLHO-10)—that was applied, in their validation study, to 51 German hospitals. This instrument has proven to have satisfactory reliability and validity, and it provides a useful way to assess the degree to which healthcare organizations help patients to navigate, understand, and use information and services [22]. Moreover, in the same study, the authors found a positive correlation between scores on the HLHO-10 and patients’ perceived adequacy of information provided by hospitals, as a measure of quality of care related to relational aspects. These results have been confirmed in a study conducted in Turkey [23], in which OHL, measured using the HLHO-10, seems to be a significant determinant of patient satisfaction. The HLHO-10 has been also used in Arkansas (USA) in a study conducted to assess healthcare practices at an academic health center [24].

To the best of our knowledge, to date, no robust studies have been conducted to assess whether the quality of care perceived by the managers and clinicians-in-chief of healthcare organizations is related with OHL [25].

In fact, the provider’s view about service quality is crucial for two reasons.

On the one hand, the patient’s difficulty in evaluating some of the most important dimensions of a healthcare service, such as physicians’ technical competence, leads some authors [26] to consider patients’ views as not significant in this sector, since they are mostly based on aspects far from the effectiveness of the care process. Instead, the importance of healthcare workers’ perspectives has been confirmed by different studies [27], showing that providers are in a good position to assess the service quality: they are in direct contact with patients and therefore more able to observe the results of the care process [28]. Furthermore, providers should know the organizational dynamics and techniques necessary to support the delivery process.

On the other hand, the providers can play an important role in increasing HL [29]. In general, providers recognize that fully engaging patients is crucial in increasing service quality [30]. Thus, providers should assign an important role to HL, and more in general to the environment that supports HL practices, even if no robust empirical evidence is provided by the literature about their perceptions regarding the effects of HL on service quality. The studies that analyze this relationship have, in fact, a small sample size and the authors themselves recognize the impossibility of making causal inferences from these data [31].

In conclusion, providers can represent privileged witnesses to assess the quality of the services that they provide, as well as their relationships with OHL.

The aims of this study were to validate the HLHO-10 in the Italian language; to measure the degree of implementation of the 10 attributes of HLHOs in a sample of hospitals placed in Tuscany; and to assess the association between the degree of implementation of the 10 attributes of HLHOs and the quality of care, related to relational aspects, perceived by the healthcare managers of the sample of Tuscan hospitals. The inclusion of different types of hospitals (public teaching, local public, and accredited private hospitals) allowed providing a comprehensive point of view about the healthcare system, as well as bringing out some key elements of the different types of healthcare organizations.

## 2. Materials and Methods

This study, according to the principles expressed in the Helsinki Declaration, was conducted among healthcare managers working for all of the local public hospitals placed in the Tuscan Health System named “Azienda Sanitaria Toscana Centro”, which comprises the three provinces of Firenze, Prato, and Pistoia (1,628,345,000 inhabitants, equal to 43.6% of the Tuscan population), and the public teaching hospital, placed in Florence. Moreover, healthcare managers working for accredited private hospitals placed in the same area and belonging to the Italian Association for Private Hospitals (AIOP, one of the most representative associations for private hospitals in Italy) were involved as well.

These three types of healthcare organizations (public teaching hospital, local public hospital, and accredited private hospital) represent a comprehensive point of view about the healthcare system, as all of them can provide services inside the Italian National Health Service.

### 2.1. Study Design and Population

This was a cross-sectional study conducted in 2017–2019. Data were collected using a self-administered questionnaire, which the responders could answer by filling in the paper or using the digital format, as preferred.

The inclusion criteria foresaw the participation of the following managers: medical managers, non-medical healthcare managers, and technical/administrative managers. The managers worked in the above-cited settings: 1public teaching hospital; 13 local public hospitals; and 6 accredited private hospitals affiliated to AIOP (Table 1). The reduced number of private hospitals depends on the fact that these structures are, in Tuscany, much lower in number and generally smaller than the public hospitals.

The only exclusion criterion was voluntary non-participation in the study.

We asked a representative of each hospital (the Chief Medical Officer) for a comprehensive list of managers working in the structure.

Then, the questionnaire was sent to all the 699 eligible healthcare managers, followed by a presentation letter explaining the details and the aims of the study. Two reminders in the following 15 days were sent to the selected professionals. The questionnaire was fully completed by 405 healthcare managers.

### 2.2. Questionnaire

The questionnaire included three sections. In the first section, the following variables were collected: working hospital, age, sex, number of years of work in healthcare settings, number of years in unit (working in that structure, as a manager or not), number of years with a leadership/executive role, role currently played in the structure, and number of coordinated staff units (0; 1–15; 16–30; >30).

The perceived quality of the service provided by the healthcare structure was also specifically investigated in a second section by using the following question: “Is the unit that you coordinate offering an adequate quality service about non-clinical aspects related to the relationship with the patient?” [32]. The question was answered using an 8-point Likert scale, which included the following options: 1 (completely disagree), 2 (partially disagree), 3 (slightly disagree), 4 (undecided), 5 (slightly agree), 6 (partially agree), 7 (completely agree), and NA (not applicable) [26].

The third and last section consisted of the Italian version of the HLHO-10 questionnaire, developed and validated by Kowalski et al. in the German language and published in English [22]. It is composed of a Likert scale survey with 10 items that investigate the 10 attributes of HLHO [18,22]. The Likert scale was the same as the one used in the previous section.

### 2.3. Statistical Analysis

#### 2.3.1. Validation of the HLHO-10 Questionnaire

As Kowalski et al. [22], we assessed the validity of the 10-item instrument, assuming that the 10 items can assess the degree to which healthcare organizations can be considered an HLHO. In order to strengthen the translation process in the Italian language, two independent pairs of translators (i.e., two native English speakers and two native Italian speakers) autonomously translated and later proofread the results of the back-translation process. The research team compared the two back-translated versions, the original one, and the two versions in the Italian language in order to assess and verify any discrepancies emerging from the process. At the end of the process, a shared Italian version of the HLHO-10 questionnaire was obtained [33]. After this phase, the entire version of the questionnaire was shared with a small group of healthcare managers to evaluate its comprehensibility. Specifically, a brainstorm phase within a core group of experts was conducted, ending with a consensus on the final form of the entire questionnaire.

Then, the entire questionnaire was pretested in two of the local public hospitals (Prato and Pistoia) of the Azienda Sanitaria Toscana Centro in 2017. The pretesting phase showed good acceptance by the respondents, as well as a good internal consistency (Cronbach’s alpha = 0.867) [34].

A Principal Component Analysis (PCA) with varimax rotation was performed to verify the existence of latent factors in the HLHO instrument.

Then, for each component, Cronbach’s alpha was calculated to assess the internal consistency, that is, the degree to which the respondents answered consistently.

#### 2.3.2. Descriptive Analysis and Association between the Degree of Implementation of the 10 Attributes of HLHOs and the Quality of Care

A descriptive analysis was performed by synthetic measurements (mean, median, or percentages). For all of the numeric variables, normality was assessed using the Shapiro–Wilk test.

For each questionnaire, the median score of the 10 items of the HLHO-10 was calculated as the final score of the scale to produce a synthetic value to assess the perception of the professionals toward the promotion/application of HL culture within the hospital organization.

The Spearman coefficient was calculated to detect the correlation between the HLHO median score (the HLHO score) with its individual items, the item on perceived quality, and some characteristics of the managers and hospitals involved (number of hospital discharges, years of work in the structure, in the direction and in the health system, age of the professional).

Student *t*-test, ANOVA, or the analogous non-parametric tests were conducted to examine the association between age, years in unit, the HLHO score, the “perceived quality item” score, and the type of hospital (local public, accredited private, public teaching). The association between categorical data was assessed using the Fisher exact test.

For each type of hospital (local public, accredited private, public teaching), multivariate linear regression analysis was performed considering the perceived quality item as the outcome (dependent) variable and the variables significantly associated at the univariate analysis as the independent variables. For comparison, a similar analysis was conducted for all data, without stratifying by type of hospital. A backward stepwise procedure was applied.

For all of the analyses, a *p*-value of 0.05 was considered significant. The statistical analyses were conducted using Stata 14 SE (StataCorp, College Station, Texas, USA) and IBM SPSS 24 (IBM, Armonk, NY, USA).

## 3. Results

### 3.1. Sample Characteristics

A total number of 405 managers answered the questionnaire (mean compliance: 54.9%. Range: 52.1–100%). Four of them were excluded from the analysis due to an excess of missing items (more than seven missing items) in the HLHO-10 questionnaire. The analyses were then conducted on the remaining 401 questionnaires. Among them, 235 (58.6%) were working in local public hospitals, 45 (11.2%) in accredited private hospitals, and 121 (30.2%) in the public teaching hospital.

Table 2 describes the respondents’ characteristics. There were more women than men (54.1% vs. 45.9%) in every healthcare setting considered in this study. The mean age was 55.6 years (Standard Deviation (SD)= 7.41) and the mean years in unit was 10.92 years (SD = 7.37). Age was significantly different by type of hospital (*p* = 0.001), with workers of accredited private hospitals younger than the others.

Most of the included managers were medical managers (216, 56%) and non-medical healthcare managers (154, 39.9%). The role of the respondents was significantly (*p* < 0.001) different by type of hospital: in the public teaching hospital, the percentage of medical managers was higher than those of the others, while the percentages of non-medical healthcare managers and technical or administrative managers were lower.

### 3.2. Validity and Reliability of the HLHO-10 Questionnaire

The percentage of missing or non-applicable values by item varied from 1%—item 8: “Are efforts made to ensure that patients can find their way at your hospital without any problems (e.g., direction signs, information staff)?”—to 5.7%—item 9: “Do you communicate openly and comprehensibly at your hospital to your patients in advance about the costs which they themselves have to pay for treatment (e.g., out-of-pocket payments)?”. The majority of the items (*N* = 9) presented a percentage of missing values lower than 5% (Table 3). Moreover, 83% of the respondents fulfilled all the 10 items of the HLHO-10 (0% of missing or non-applicable data).

PCA with varimax rotation revealed three dimensions that explain 65% of the total variance. The first factor explained 41.5% of total variance (eigenvalue = 4.15), the second factor added an additional 13.4% (eigenvalue = 1.34), and the third factor added an additional 10.1% (eigenvalue = 1.01), suggesting the existence of three dominant latent factors. Items 1, 2, and 10 were included in the first dimension; these items are the only ones that explicitly contain the terms “health literacy”, specifically focusing on this topic at the management level. Items 3, 4, 5, and 8 were included in the second dimension: these items regard information and communication between professionals and patients. Finally, items 6, 7, and 9 were included in the third dimension: these items refer to the navigation of patients inside the hospital and to the communication of the costs for treatments.

For the first dimension, the Cronbach’s alpha was 0.819, for the second dimension, it was 0.719, while for the third dimension, it was 0.758. These findings suggest satisfactory internal reliability.

### 3.3. Validity and Reliability of the HLHO-10 Questionnaire

The scores of each item of the HLHO-10 questionnaire and the HLHO score, as well as the perceived quality item, were not normally distributed.

All of the 10 items of the HLHO-10 were significantly correlated (rho values between 0.204 and 0.636; *p* < 0.05). Moreover, each of them was significantly correlated with the perceived quality item (rho values between 0.308 and 0.426; *p* < 0.05) (Table 4).

Descriptive analysis of the items of the HLHO-10 questionnaire, the HLHO score, and the perceived quality item for local public, accredited private, and public teaching hospitals are displayed in Table 3.

In the whole sample, all of the items, with the exception of number 10 of the HLHO-10 questionnaire, present mean and median values equal to or higher than 4 (“undecided”), indicating that the respondents tend to agree with the statements (from “slightly” to “completely”), showing a good perception about HLHO and quality attributes in their organizations.

Item 4 of the HLHO-10 questionnaire (“Is individualized health information used at your hospital (e.g., different languages, print sizes, braille)?”) had the highest mean value (5.97 ± 1.54), in particular for accredited private hospitals (6.33 ± 1.22), showing that healthcare providers take great care of the personalization of communications.

Item 2 (“Is the topic of health literacy considered in quality management measures at your hospital?”) and item 10 (“Are employees at your hospital trained on the topic of health literacy?”) had the lowest means (respectively, 4.09 ± 2.07 and 3.86 ± 2.14).

For most of the items (1, 3, 4, 6, 7, 9, 10), the distribution of the score was significantly different by type of hospital (*p* < 0.05): accredited private hospitals presented higher values in each item, except for item 9—“Do you communicate openly and comprehensibly at your hospital to your patients in advance about the costs which they themselves have to pay for treatment (e.g., out-of-pocket payments)?”—for which the lowest value was observed.

Both the HLHO score and the perceived quality score were significantly associated with the type of hospitals (*p* = 0.016 and *p* = 0.015, respectively): accredited private hospitals had higher HLHO scores (mean value: 5.94 ± 1.16 vs. 5.27 ± 1.58 and 5.37 ± 1.44) and perceived quality score (6.24 ± 1.05 vs. 5.55 ± 1.63 and 5.88 ± 1.45), followed by public teaching hospitals, and then local public hospitals. Otherwise, HLHO scores and the perceived quality scores were not significantly different by sex or professional role.

A correlation analysis was conducted between the HLHO score, the perceived quality score, the number of years of working experience in chief, the age of the respondents, and the number of hospital admissions in 2018 in the working hospital. The HLHO score was significantly correlated with perceived quality (rho = 0.503; *p* < 0.01), and with the number of hospital admissions in the working hospital (rho = −0.125; *p* = 0.013). Perceived quality was significantly correlated with years of working experience in chief as well (rho = 0.163; *p* = 0.002).

Multivariate linear regression models were conducted to investigate the relationship between the perceived quality (outcome variable) and the correlated or associated variables (HLHO score, number of hospital admissions in 2018, number of years of working experience in chief) by type of hospital and in the whole sample. In Table 5, the original and the final models are reported (the final models are those including the variables that maintained a statistically significant association with the outcome variable applying the backward stepwise procedure). Specifically, the number of hospital admissions in 2018 was excluded for each type of hospital, while the number of years of working experience in chief was excluded both for accredited private and for the teaching public hospitals. The analysis shows that the HLHO score has a significant role in predicting the perceived quality for each type of investigated hospital and for the whole sample: perceived quality increases with the increase of HLHO score, with the highest coefficient for local public hospitals (beta = 0.596).

## 4. Discussion

To the best of our knowledge, this is the first Italian research that measures the OHL by means of a specific tool and that compares the OHL with the perceived quality about non-clinical aspects on the part of the healthcare providers.

The aims of this study were threefold. Concerning the first aim (i.e., to validate the HLHO-10 in the Italian language), the results show that the measurement tool presents good psychometric characteristics, so it could be considered valid. As regards PCA, differently from what was observed by Kowalski [22], three dimensions emerge, suggesting a multidimensionality of the tool. The three dimensions are mainly linked, respectively, to the explicit reference to the HL issue in the policy of the hospitals, to the communication between professionals and patients, and to the navigation of the patients inside the hospitals, including also the communication of the costs for treatments. These results, confirmed also by the correlation analysis, are in line with the conceptualization of OHL, in which several domains are identified [20,35,36]. The difference in the number of dimensions is probably due to the fact that our study has considered a different target, compared to the research of Kowalski et al. In their work, the selected structures are all similar and have specific care recipients: breast cancer center hospitals and people who suffered from breast cancer. This relative homogeneity could have influenced the results of the PCA analysis.

Moreover, the percentage of missing values is lower than 5% for most of the items, which provides evidence of respondents’ acceptance and understanding of the items and could be considered as a proxy of instrument sensitivity [37].

Considering the answers to each item of the HLHO and the final score, it emerges that, according to the perception of the providers, the investigated hospitals are quite health literate, although some areas need improvement and some differences between types of hospitals are significant. Specifically, the areas with worse situations seem to be those related to the explicit integration of heath literacy into management practices and those related to the training of the employees on the topic of health literacy. On the other hand, higher scores are reported for the aspects that deal with information and communication between professionals and patients. These data seem to indicate that, in the investigated sample, the HL principles are not explicitly incorporated into the hospital policy, but they are more broadly embedded into practices. The results are quite different from those reported in other studies in which the HLHO-10 was used: both in the German and in the Turkish hospitals, the items with lower scores were some of those related to the communication with the patients, while higher scores were described for the items related to the integration of health literacy at the managerial level [22,38].

In our sample, accredited private hospitals seem to be more health literate, followed by the teaching and then the local public hospitals, differently from those that were observed in Turkey [38].

This result could be due to the different mission that these different typologies have and the different system of relationship that characterizes them. Teaching hospitals, as well as local public hospitals, have, in fact, the primary mission to satisfy the acute healthcare needs of the patient, at the local or national level. On the contrary, accredited private hospitals have more interest in developing an HL culture, also to attract clients and establish with them a clear communication upon care.

The three healthcare systems are deeply different, and this is a significant element that can justify the dissimilar results between the considered studies. Specifically, the Turkish healthcare system is the youngest one and it has three kinds of hospitals: state-funded hospitals, which suffer from overcapacity and lack of finances, university hospitals, which have the highest standard of care, and private hospitals, that are halfway between the two. On the other hand, the German healthcare system is health-insurance-based and it is organized into: 1) competing, not-for-profit, nongovernmental health insurance funds (“sickness funds”), in the statutory health insurance (SHI) system, and 2) substitutive private health insurance.

Regarding the results of our research, we need to mention a potential limit of collecting data with self-perceived evaluation tools, that is, the bias of social desirability (or idealistic distortion); this means that people tend to give socially acceptable answers, not exactly in line with personal opinions or attitudes, but “idealized”, adequate to the rules of a socially accepted behavior or aimed to please the interviewer. To reduce the impact of this bias, some strategies are identified, including anonymous auto-compilation and computer-neutralized administration [39]. As documented, in our research, we tried to comply with these strategies. Moreover, although the measure of the perceptions can include this type of bias, the sample chosen in our study should reduce them. Indeed, our sample included only middle managers: this ensured that each item of the HLHO-10 questionnaire (even those regarding the choices made by the top management of the hospitals) was answered by the middle managers.

Another limitation is related to the study design: OHL and quality about non-clinical aspects were assessed at the same time using the same questionnaire, with a possible reduction in the variability of the responses given by the healthcare managers. Moreover, only the middle managers were involved in the study, so the significant association between OHL and quality about non-clinical aspects is the result of what is perceived by—and within—this target group. A third-party investigation, independent from this study, could be useful in order to confirm the results.

Summarizing, from our analysis, three crucial themes emerge on which healthcare organizations will have to carefully reflect. The first two require an intervention at the top management level, while the third also involves the middle management of the healthcare organizations.

The first theme underscores the importance of looking at HL as a multidimensional concept. As the literature shows, HL is increasingly recognized not just as an individual trait [4], but also as a phenomenon in which the organizational context can compensate for patients’ limited HL [6]. Indeed, when healthcare organizations support HL best practices, the quality of healthcare can be improved even if patients do not have advanced HL skills [40]. Despite this, at least as shown in our study, HL seems still to be perceived more as an element to be treated individually than as an organizational feature. This underscores the need to address the HL policy, both at the organizational level and at the system level [35]. Informal approaches to OHL—that is, when HL is left only to the initiative of single professionals that seem to characterize the analyzed hospitals—have been considered less effective than formal ones [40].

From this point of view, the top management of healthcare organizations should better integrate HL into the strategy, with the aim to better carry out their mission. The importance of this aspect is confirmed whenever the providers’ perceptions are that the top management does not devote enough attention to developing a specific organizational culture about health literacy. The poor management orientation to HL, perceived especially by providers of the public sector, may have contributed, along with other factors, to the scarce attention dedicated to training on the topic of HL. On the basis of the literature, this can represent a problem because when an organization is not HL-oriented, training on this topic cannot be omitted [41], differently from organizations oriented toward HL, where the aspect related to training on health literacy can be neglected. The importance of training seems instead better perceived by the accredited private hospitals.

A second aspect that emerges from the results of this study refers to the scarce attention assigned by the investigated hospitals to the topic of health literacy in the measure of quality management. Also, on this point, top management intervention can be important in addressing health literacy as part of continuous quality improvements [42] by means of scoring, surveys, and development of improvement plans [20].

## 5. Conclusions

This study confirms the research conducted on patients that shows that the orientation toward HL is higher in accredited private hospitals than in local public hospitals [43]. This result could emphasize the importance of distinguishing between the different missions of the hospitals that characterize the different types of organizations.

In conclusion, this study offers a new point of view on the relationship between HLHO and the quality of service: confirming the vision of the providers, there results a relationship between HLHO and the quality of service, as perceived by the middle managers. The results are in line with previous studies that assessed this feature from the patient’s point of view [23]. Therefore, in order to improve the quality of service, it seems to emerge that the OHL culture should be an integral element in organizational policies, as well as part of the toolbox of top and middle management; this latter organizational level has been calling to exert a fundamental role in organizations that are becoming bigger and bigger, where often they receive a broader delegation from top management.

HL seems to be one of the key factors to develop a managerial skill aimed to raise the quality of healthcare, as well as the responsibility and the sense of membership of the providers.

## Figures and Tables

**Table 1 ijerph-17-02508-t001:** Description of the hospitals and providers investigated (data refer to 2018).

Variables			Local Public Hospitals (*N = 13*)	Accredited Private Hospitals (*N = 6*)	Public Teaching Hospital (*N = 1*)
Hospital admissions		Mean ± SD	9879 ± 9012	2640 ± 2238	62,713
		Median	5646	1548	-
Professionals (*N* = 692)	Non-medical healthcare managers	Total (*N*)	214	26	88
		Mean ± SD by hospital	19.45 ± 11.85	5.20 ± 5.67	-
	Medical managers	Total (N)	202	13	131
		Mean ± SD by hospital	18.36 ± 9.27	4.33 ± 1.15	-
	Technical/administrative managers	Total (N)	6	4	8
		Mean ± SD by hospital	0.54 ± 0.65	2.00 ± 0.00	-

**Table 2 ijerph-17-02508-t002:** Characteristics of the sample by type of hospital.

Variables		Local Public Hospitals	Accredited Private Hospitals	Public Teaching Hospital	Total	*p* *
Sex (*N*; %) *(45 missing)*	Females	122; 51.9%	25; 55.6%	70; 57.9%	217; 54.1%	0.310
	Males	113; 48.1%	20; 44.4%	51; 42.1%	184; 45.9%	
Role (*N*; %) *(15 missing)*	Non-medical healthcare managers	116; 52.3%	26; 60.5%	12; 51.2%	154; 39.9%	<0.001
	Medical managers	96; 43.2%	13; 30.2%	107; 88.4%	216; 56.0%	
	Technical/ administrative managers	10; 4.5%	4; 9.3%	2; 1.7%	16; 4.1%	
Age (years) *(9 missing)*	Min	38	34	31	31	0.001
	Max	70	70	71	71	
	Mean	55.8	51.5	56.8	55.6	
	Median	56.6	50	58	56	
	Standard deviation	6.41	9.95	7.58	7.41	
Number of years in unit (years) *(34 missing)*	Min	0	1	0	0	0.115
	Max	30	30	38	38	
	Mean	10.04	12.45	10.89	10.92	
	Median	9.5	11.5	10	10	
	Standard deviation	6.9	7.27	7.33	7.37	

***** For “sex” and “role”: Fisher exact test; for “age” and “number of years in unit”: Kruskal–Wallis test for independent samples.

**Table 3 ijerph-17-02508-t003:** Descriptive analysis of the items of the 10-item questionnaire for Health-Literate Healthcare Organization (HLHO-10), the HLHO score, and the perceived quality item (QUAL1) for the whole sample and by type of hospital.

		% Missing	Mean	Median	SD	*p* *
**HLHO1—Is the management at your hospital explicitly dedicated to the subject of health literacy (e.g., mission statement, human resources planning)?**	Local public hospitals	1.3	4.37	5	2.0	<0.001
	Accredited private hospitals	2.2	5.61	6	1.47	
	Public teaching hospital	1.7	4.36	5	2.04	
	Total	1.5	4.50	5	1.994	
**HLHO2—Is the topic of health literacy considered in quality management measures at your hospital?**	Local public hospitals	2.6	4.05	4	2.07	0.081
	Accredited private hospitals	2.2	4.77	5	1.74	
	Public teaching hospital	3.3	3.91	4	2.15	
	Total	2.7	4.09	4	2.07	
**HLHO3—Is health information at your hospital developed by involving patients?**	Local public hospitals	0.9	5.03	6	1.93	0.025
	Accredited private hospitals	0	5.89	6	1.33	
	Public teaching hospital	12.4	5.15	6	1.87	
	Total	4.2	5.16	6	1.875	
**HLHO4—Is individualized health information used at your hospital (e.g., different languages, print sizes, braille)?**	Local public hospitals	2.6	5.74	6	1.62	<0.001
	Accredited private hospitals	0	6.33	7	1.22	
	Public teaching hospital	9.9	6.28	7	1.39	
	Total	4.5	5.97	7	1.54	
**HLHO5—Are there communication standards at your hospital which ensure that patients truly understand the necessary information (e.g., translators, allowing pauses for reflection, calling for further queries)?**	Local public hospitals	1.7	5.76	6	1.41	0.620
	Accredited private hospitals	0	5.87	6	1.32	
	Public teaching hospital	12.4	5.75	6	1.67	
	Total	4.7	5.77	6	1.48	
**HLHO6—Are efforts made to ensure that patients can find their way at your hospital without any problems (e.g., direction signs, information staff)?**	Local public hospitals	0.9	5.44	6	1.72	0.027
	Accredited private hospitals	2.2	5.93	6	1.42	
	Public teaching hospital	0.8	5.25	6	1.72	
	Total	1.0	5.44	6	1.70	
**HLHO7—Is information made available to different patients via different media at your hospital (e.g., three-dimensional models, DVDs, picture stories)?**	Local public hospitals	0.4	5.30	6	1.76	0.049
	Accredited private hospitals	11.1	5.90	7	1.43	
	Public teaching hospital	3.3	5.33	6	1.68	
	Total	2.5	5.37	6	1.71	
**HLHO8—Is it ensured that the patients have truly understood everything, particularly in critical situations (e.g., medication, surgical consent), at your hospital?**	Local public hospitals	1.7	5.73	6	1.48	0.914
	Accredited private hospitals	2.2	5.64	6	1.57	
	Public teaching hospital	9.9	5.69	6	1.53	
	Total	4.2	5.71	6	1.50	
**HLHO9—Do you communicate openly and comprehensibly at your hospital to your patients in advance about the costs which they themselves have to pay for treatment (e.g., out-of-pocket payments)?**	Local public hospitals	0	5.28	6	1.67	<0.001
	Accredited private hospitals	4.4	4.14	4	1.37	
	Public teaching hospital	17.4	5.32	6	1.82	
	Total	5.7	5.16	6	1.72	
**HLHO10—Are employees at your hospital trained on the topic of health literacy?**	Local public hospitals	0.9	3.58	4	2.05	<0.001
	Accredited private hospitals	0	6.18	7	1.15	
	Public teaching hospital	3.3	3.54	3	2.09	
	Total	2.5	3.86	4	2.14	
**HLHO score**	Local public hospitals	-	5.27	6	1.58	0.016
	Accredited private hospitals	-	5.94	6	1.16	
	Public teaching hospital	-	5.37	6	1.44	
	Total	-	5.37	6	1.51	
**QUAL1—Is the unit that you coordinate offering an adequate quality service about non-clinical aspects related to the relationship with the patient?**	Local public hospitals	3.8	5.56	6	1.63	0.015
	Accredited private hospitals	0	6.24	7	1.05	
	Public teaching hospital	6.6	5.88	6	1.45	
	Total	4.2	5.73	6	1.54	

***** Kruskal–Wallis test for independent samples.

**Table 4 ijerph-17-02508-t004:** Spearman correlation analysis. For each rho value, *p* < 0.05.

	QUAL1	HLHO1	HLHO2	HLHO3	HLHO4	HLHO5	HLHO6	HLHO7	HLHO8	HLHO9
**QUAL1**	1									
**HLHO1**	0.426	1								
**HLHO2**	0.341	0.636	1							
**HLHO3**	0.376	0.297	0.258	1						
**HLHO4**	0.387	0.305	0.221	0.503	1					
**HLHO5**	0.312	0.208	0.204	0.384	0.455	1				
**HLHO6**	0.316	0.404	0.369	0.280	0.314	0.280	1			
**HLHO7**	0.319	0.346	0.385	0.297	0.290	0.227	0.718	1		
**HLHO8**	0.315	0.278	0.268	0.393	0.368	0.379	0.316	0.312	1	
**HLHO9**	0.308	0.330	0.377	0.226	0.252	0.264	0.388	0.371	0.367	1
**HLHO10**	0.440	0.557	0.520	0.285	0.262	0.233	0.307	0.324	0.254	0.221

**Table 5 ijerph-17-02508-t005:** Multivariate linear regression analysis: original (A) and final (B) models by type of hospital and in the whole sample. Dependent variable: perceived quality. Data are adjusted for age and role.

**A.** **ORIGINAL MODELS**	**Beta**	**Standard Error**	***t***	***p* > | t |**	**[95% Conf. Interval]**	
1. Local public hospitals (*N* = 203)						
years of working experience in chief	0.043	0.016	2.664	0.008	0.011	0.075
HLHO score	0.589	0.063	9.319	<0.001	0.465	0.714
number of hospital admissions in 2018	-0.00001	0	-0.918	0.36	0	0
2. Accredited Private hospitals (*N* = 45)						
years of working experience in chief	-0.006	0.03	-0.183	0.856	-0.067	0.056
HLHO score	0.381	0.193	1.972	0.057	-0.012	0.774
number of hospital admissions in 2018	-0.00002817	0	-0.272	0.787	0	0
3. Public teaching hospital (*N* = 113)						
years of working experience in chief	0.009	0.022	0.393	0.695	-0.035	0.052
HLHO score	0.238	0.092	2.597	0.011	0.056	0.42
4. All (*N* = 401)						
years of working experience in chief	0.024	0.011	2.132	0.034	1.7	4.303
HLHO score	0.489	0.049	10.051	<0.001	0.002	0.046
number of hospital admissions in 2018	0.000004	0	1.146	0.252	-0.101	0.061
**B. FINAL MODELS**	**Beta**	**Standard Error**	***t***	***p*** **> | t |**	**[95% Conf. Interval]**	
1. Local public hospitals (*N* = 203)						
years of working experience in chief	0.042	0.016	2.628	0.0089	0.011	0.074
HLHO score	0.604	0.061	9.88	<0.001	0.484	0.725
2. Accredited private hospitals (*N* = 45)						
HLHO score	0.346	0.133	2.6	0.013	0.077	0.616
3. Public teaching hospital (*N* = 113)						
HLHO score	0.242	0.091	2.654	0.009	0.061	0.422
4. All (*N* = 401)						
years of working experience in chief	0.024	0.011	2.117	0.035	0.002	0.046
HLHO score	0.487	0.049	10.004	<0.001	0.391	0.582

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
