# Peer review of "Health-Literate Healthcare Organizations and Quality of Care in Hospitals: A Cross-Sectional Study Conducted in Tuscany"

_ijerph, 2020, doi:10.3390/ijerph17072508_

Round 1
Reviewer 1 Report
This research seeks to validate an Italian translation of the survey measure of a health literate organization developed by Kowalski and colleagues against a measure of perceived quality. Measuring how health literate hospitals are is an important issue and the authors are well acquainted with the HLO literature in this field. However, their study suffers from a major design flaw.
Unlike the two previous HLHO validation studies, which both used patient reported data and/or patients safety measures, this study relies on reports of perceived quality by the same managers that complete the HLHO survey items. The measure of perceived quality was validated against patient safety issues in nursing. It does not appear to have been validated for use by healthcare managers for non-clinical aspects of care.
Furthermore, the data on both perceived quality and HLHO were collected as part of the same survey. The authors do not discuss the likelihood that how managers responded to the HLHO could have influenced how they responded on the measure of perceived quality. It seems likely that if you’ve just filled out 10 questions indicating that your hospital is highly (or not) health literate, you’d rate overall quality similarly.
I believe that the researchers should conduct additional analysis with another source of quality data (e.,g., quality measures that are routine reported) that are contemporaneous with the survey data collection. The fact that managers report that their hospitals are both health literate and deliver adequate quality non-clinical service is not particularly useful to the field. Research that validated the HLHO against a validated quality measure would be hugely helpful to the field.
Below are additional comments that could help the authors revise their manuscript.
- Did the pretest of the questionnaire include cognitive testing, i.e., do we know how managers were interpreting the perceived quality item.
- The unit of analysis of the Health Literate Healthcare Organization (HLHO) measure is the hospital, but the unit of analysis of the measure of perceived quality is the unit that manager coordinates. The researchers take the median HLHO scores from responding managers to create a hospital level HLHO score. However, we are not given information about how many managers per hospital responded. It is possible that some hospitals had few respondents, possibly biasing the result.
- The authors note that there were significant differences in respondents’ age and role by type of hospital, yet they don’t appear to have corrected for these differences.
- The authors don’t specify when the survey was fielded.
- The authors included questionnaires with up to 6 items missing. How was this threshold determined? How were missing data handled (e.g., was there any imputation)?
- What is the a priori rationale for constructing separate models by hospital type rather than a single model with dummy and/or interactive variables to see if HLHO varied by hospital type? I would like to see the original and final (i.e., after backward stepwise procedure) models, and results of a single model with all hospital data.
- The authors note that their results were different from other HLHO validation studies, yet don’t help us reconcile those differences. What is the hypothesis for why the authors came up with three factors while Kowalski and colleagues came up with one? Why do they think their results on types of hospitals was the converse of Hayran and Ozer’s results? Would the authors explanation of their own results, that private hospitals are competing for patients and therefore have a greater interest in developing a HL culture, not apply in Turkey?
- Lines 148-9 – What the difference between the number of years “in charge” and “with a leadership/executive role?”
- Line182 – Reference to “subject” is confusing. I was able to figure out that this referred to the hospital.
- Lines 210 – 211 – Provide p values when report significant differences.
- Table 3: The readability would be increased if the HLHO items were left justified rather than centered.
- Lines 253-3 – The authors state that the responses agreeing with HLO and quality statements shows “a good perception about HLHO and quality attributes in their organization.” The authors do not consider that since these are managers reporting about the performance of their hospital that there is a social desirability bias. Since the only data source in this study is the managers’ own reports, the authors should be cautious in interpreting the data as evidence that the hospitals are delivering health literate, high quality care.
Author Response
Point 1: Unlike the two previous HLHO validation studies, which both used patient reported data and/or patients safety measures, this study relies on reports of perceived quality by the same managers that complete the HLHO survey items. The measure of perceived quality was validated against patient safety issues in nursing. It does not appear to have been validated for use by healthcare managers for non-clinical aspects of care.
Response 1: The measure of perceived quality we use to evaluate the quality of healthcare is relative to a study whose objectives were: “to examine the effects of nurse staffing and process of nursing care indicators on assessments of the quality of nursing care” (Sochalski, 2004; p.67).
Findings of the studies show that quality of care were significantly associated with number of patients who nurses care for, the task undone and rates of unfinished care for those patients.
The reasons of the choice of the framework we used are twofold.
On the one hand, some of the items included in the survey that the study used to evaluate the variable “task undone” (that have a strong association with the quality of nursing care) measure non-clinical aspect of the care, such as: “patient teaching and counseling, documenting patient problems and interventions, and discharge planning”.
On the other hand, the use of a single item to evaluate the quality of the care is considered important, as authors state because: “a single item could capture not only a broad set of attributes, but also the more intangible aspects of the care” such as the non-clinical aspect or relational concerns that are the focus of our study. In conclusions, the framework we used to measure of perceived quality, was validated for use by healthcare managers also for non-clinical aspects of care as the same author stated.
Point 2: Furthermore, the data on both perceived quality and HLHO were collected as part of the same survey. The authors do not discuss the likelihood that how managers responded to the HLHO could have influenced how they responded on the measure of perceived quality. It seems likely that if you’ve just filled out 10 questions indicating that your hospital is highly (or not) health literate, you’d rate overall quality similarly. I believe that the researchers should conduct additional analysis with another source of quality data (e.,g., quality measures that are routine reported) that are contemporaneous with the survey data collection. The fact that managers report that their hospitals are both health literate and deliver adequate quality non-clinical service is not particularly useful to the field. Research that validated the HLHO against a validated quality measure would be hugely helpful to the field.
Response 2: Thanks for the comment. The aim of this study, that was the first one conducted in Italy, was to measure OHL and to assess the relationship between the OHL and the perceived quality about non-clinical aspects on the part of the healthcare providers. For this reason, the study design was that of a cross-sectional study, conducted at the individual level, i.e. administering to each healthcare professional satisfying the inclusion criteria the questionnaire developed to measure, at the same time, OHL and perceived quality about non-clinical aspects. The study design respects the anonymity and the inability to trace the person who completed the questionnaire (which has been a specific condition requested to the participating structures and to the Ethic Committee), so that it is not possible to link each questionnaire collected in this survey to other ones which could be collected in other surveys. For this reason, the only way to match the HLHO-10 questionnaire individual responses to the perceived quality about non-clinical aspects was to include both measures in the same questionnaire. This study design allows us to collect other individual data, and to include these variables in the analysis in order to test their association with OHL and with the perceived quality about non-clinical aspects. However, we are aware that it could be very useful to confirm our results using a third-party investigation, independent from this study. For this reason, we are planning an ecological study, conducted at the hospital-level, to investigate the relationship between the scores collected in this study (HLHO and the quality of care scores) and administrative data related to quality of care, routinely collected. We hope to realize the ecological study as soon as possible in order to better investigate the role of OHL in the quality of care provided by hospitals. In the discussion this aspect has been added.
Point 3: Below are additional comments that could help the authors revise their manuscript.
Did the pretest of the questionnaire include cognitive testing, i.e., do we know how managers were interpreting the perceived quality item.
Response 3: Unfortunately, cognitive testing was not included in the pretest of the questionnaire. However, before being administered to the pretesting sample, the questionnaire was shared with a small group of healthcare managers to evaluate its comprehensibility. Specifically, a brainstorm phase within a core group of experts was conducted, ended with a consensus on the final form of the entire questionnaire. This information has been added in the manuscript.
Point 4: The unit of analysis of the Health Literate Healthcare Organization (HLHO) measure is the hospital, but the unit of analysis of the measure of perceived quality is the unit that manager coordinates. The researchers take the median HLHO scores from responding managers to create a hospital level HLHO score. However, we are not given information about how many managers per hospital responded. It is possible that some hospitals had few respondents, possibly biasing the result.
Response 4: The description of the percentage of respondents by hospital has been added in the results as a range that varies from 52,1% to 100%.
Point 5: The authors note that there were significant differences in respondents’ age and role by type of hospital, yet they don’t appear to have corrected for these differences.
Response 5: Although age and role of the respondents differ by type of hospital, they were not significantly associated with either the HLHO score or the perceived quality scores. For this reason, they were not included in the regression model, as for the other variables without significant association with the HLHO score or the perceived quality scores. Nonetheless, to take into account this comment, we have modified the regression analysis including also age and role of the respondents, as adjustment variables. Please, note that the results obtained adjusting for these variables are consistent with the previous one.
Point 6: The authors don’t specify when the survey was fielded.
Response 6: This information has been added
Point 7: The authors included questionnaires with up to 6 items missing. How was this threshold determined? How were missing data handled (e.g., was there any imputation)?
Response 7: The threshold of 6 item was arbitrarily determined, as the result of a consensus within the research group. Data were analyzed without assumptions or imputation for missing data. If it could be useful, we can modify the analysis including a method for imputation of missing data. Please, note that we have grouped together missing data to “non-applicable” response, since, in our opinion, they are both useful in assessing the validity of the questionnaire, although missing data may occur to different reason from “non-applicable” response. As described in the results, the percentage of missing plus non-applicable data by item was low. Moreover, only a few questionnaires (N=4) presented an excess of missing items (more than seven missing items); this data suggest that missing or non-applicable data are not concentrated into clusters. However, if it could be useful, we can add a deeper descriptive analysis of missing and non-applicable data by item and by respondent. Finally, as suggested by the Referee n. 4, the percentage of questionnaire without missing plus non applicable data at the HLHO-10 has been added, and it is equal to 83%.
Point 8: What is the a priori rationale for constructing separate models by hospital type rather than a single model with dummy and/or interactive variables to see if HLHO varied by hospital type? I would like to see the original and final (i.e., after backward stepwise procedure) models, and results of a single model with all hospital data.
Response 8: The study has been conducted in three different types of healthcare organizations (public teaching hospital, local public hospital, and accredited private hospital) in order to give a comprehensive point of view about the healthcare system, as all of them can provide services inside the Italian National Health Service (see the Methods section). Moreover, a previous study conducted in Turkey (Hayran, 2018) has shown interesting differences among different types of hospital. For this reason, the majority of our descriptive analyse as well as the regression analysis have been performed by type of hospital (Tables 3 and 5). To better highlight this aspect, the aim of the study has been changed, adding the following statement: “The inclusion of different types of hospitals (public teaching, local public, and accredited private hospitals) allowed to provide a comprehensive point of view about the healthcare system, as well as to bring out some key elements of the different types of healthcare organizations”. Nonetheless, as requested, we have added, for comparison, a new regression model including all the data. Moreover, the original results (i.e. before backward stepwise procedure) have been added for each model. Please, note that the results are confirmed also including the analysis for the entire sample and adjusting for age and role.
Point 9: The authors note that their results were different from other HLHO validation studies, yet don’t help us reconcile those differences. What is the hypothesis for why the authors came up with three factors while Kowalski and colleagues came up with one? Why do they think their results on types of hospitals was the converse of Hayran and Ozer’s results? Would the authors explanation of their own results, that private hospitals are competing for patients and therefore have a greater interest in developing a HL culture, not apply in Turkey?
Response 9: As regards to the PCA analysis, this is what emerges from our measured results: we cannot forced all the three components in only one as for Kowalski, as appeared by data analysis, and we have explained this in the text, attributing each item to a specific component, which seems also appropriate from a theoretical point of view. What can seem more probable in assessing the difference in the number of components is attributable to the fact that our study has considered a different target respect to the German and the Turkish researches: in the study of Kowalski et al., the selected structures are all similar and regarded a specific casuistry of patients, that is, breast cancer center hospitals and people who suffered from the same disease. This relative homogeneity could, and effectively should, influence the results. In the study of Hayran and Ozer the investigation regarded mainly patient satisfaction and not the perceived quality of care on the part of the managerial staff, so that it cannot be comparable since it explores a different point of view. Regarding the face-to-face interviews conducted with the staff management of the three Turkish hospitals, the number is very dissimilar from ours: 18 managers against 405. It is not the same kind of analysis and in fact the aims of the Turkish colleagues interested also patients’ HL and satisfaction, which are not among our aims.
Finally, a fundamental difference is due to the fact that the three Healthcare Systems are profoundly different: the Turkish one is the youngest and subdued very complex changes until the recent HTP - Health Transformation Programme - in 2003. There are three types of hospitals throughout Turkey: State-funded hospitals, which suffer form over capacity and lack of finances; University Hospitals, which have the highest standard of care out of all three of hospital types and boast highly skilled personal, and private hospitals, which poses in the middle. The results reported by Hayran and Ozer are the mirror of this kind of organization.
The German Healthcare System follows the so-called Bismarck-like model, that is a Health insurance-based system in which insurance is mandatory for all citizens and permanent residents of Germany. It is provided by two systems, namely: 1) competing, not-for-profit, nongovernmental health insurance funds (“sickness funds) in the statutory health insurance (SHI) system; and 2) substitutive private health insurance (PHI). States own most university hospitals, while municipalities play a role in public health activities and own about half of all hospital beds.
The Italian Healthcare System is, instead, based on a Beveridge-like model, and has benne founded in 1978 and partially modified with the 2004 Constitutional Reform: the reform has given some organizational elements and the way of service provisioning to regional governments, but maintains the structure of a unique National Healthcare System. In the majority of Italian regions, the full range of healthcare services is provided by public organizations. This is particularly true for Tuscany – the territory in which our research is based - where private structures are few with most of them providing services in accordance with and refunded by the regional healthcare system: this cannot be ignored in the attitudes of the participating private organizations towards health literacy, considered also as a key to gain reputation and “clients”.
Point 10: Lines 148-9 – What the difference between the number of years “in charge” and “with a leadership/executive role?”
Response 10: “Years in charge” refers to the number of years of work in the same structure in which we are conducting our interview, while “years with a leadership/executive role” refers to the number of years working as manager. This information has been added in the manuscript where “in charge” appears for the first time.
Point 11: Line182 – Reference to “subject” is confusing. I was able to figure out that this referred to the hospital.
Response 11: The term has been checked in the whole manuscript and changed where its reference was confusing.
Point 12: Lines 210 – 211 – Provide p values when report significant differences.
Response 12: The p value was added in the text when significant differences are described
Point 13: Table 3: the readability would be increased if the HLHO items were left justified rather than centered.
Response 13: The table has been changed according to the suggestion.
Point 14: Lines 253-3 – The authors state that the responses agreeing with HLO and quality statements shows “a good perception about HLHO and quality attributes in their organization.” The authors do not consider that since these are managers reporting about the performance of their hospital that there is a social desirability bias. Since the only data source in this study is the managers’ own reports, the authors should be cautious in interpreting the data as evidence that the hospitals are delivering health literate, high quality care.
Response 14: The social desirability bias has been discussed as a limit of the study. However, we would like to argue that, although the measure of the perceptions can include this type of bias, the sample chosen in our study should reduce them. In fact, our sample includes only middle managers. So, our results are relative to the perceptions of middle managers also when the items of the HLHO-10 questionnaire regard the choices made by the top management of the hospitals. Specifically, the items of the HLHO-10 questionnaire that are referred to the choices made of the top management of the hospitals are the following:
HLHO1—Is the management at your hospital explicitly dedicated to the subject of health literacy (e.g., mission statement, human resources planning)?
HLHO2—Is the topic of health literacy considered in quality management measures at your hospital?
HLHO10—Are employees at your hospital trained on the topic of health literacy?
Reviewer 2 Report
Dear Authors,
I assess your paper highly. The topic is very important in the aspect of improving the quality of medical services.
You described the questionnaire and your methods perfectly.
I'll be grateful if you might consider mentioning what kind of software you have used.
Was is SPSS or Statistica? Maybe R or something different?
It is very interesting for the readers.
Author Response
Point 1: I assess your paper highly. The topic is very important in the aspect of improving the quality of medical services. You described the questionnaire and your methods perfectly. I'll be grateful if you might consider mentioning what kind of software you have used. Was is SPSS or Statistica? Maybe R or something different?
Response 1: Thanks for the comments. For the analysis, Stata 14 SE and IBM SPSS 24 was used. This information has been added to the manuscript.
Reviewer 3 Report
Dear authors,
Your paper is very interesting. However, you must specify some items before accepting it:
MATERIALS AND METHODS
2.3.1. Validation of the HLHO-10 questionnaire
- Authors should provide evidence of validity of the scale.
REFERENCES
Many bibliographies are obsolete. The bibliographic citations used are more than 5 years old (56,1%). The authors must update and arrange the bibliography.
There are references that have errors. Authors should review the citation.
Author Response
Point 1: Dear authors, your paper is very interesting. However, you must specify some items before accepting it:
MATERIALS AND METHODS
2.3.1. Validation of the HLHO-10 questionnaire
- Authors should provide evidence of validity of the scale.
Response 1: In the 2.3.1 section, more details regarding the pretesting phase were added.
Point 2: REFERENCES
Many bibliographies are obsolete. The bibliographic citations used are more than 5 years old (56,1%). The authors must update and arrange the bibliography.
There are references that have errors. Authors should review the citation.
Response 2: The references have been checked and updated, where possible. Please, note that some of the older references cannot be updated since they are key documents for the research issue.
Reviewer 4 Report
Dear Authors,
I read with pleasure and great interest the work devoted to the extremely important issue which is quality of care. It analyzes in an innovative way the phenomenon taking into consideration non-clinical conditions. Moreover, unlike most studies in the field of quality of care that focus on recipients of services, it focuses on the persons responsible for arranging services, i.e. the management.
Sufficient introduction, the authors could, however, provide accurate information in which countries the tool was used. Work methodology was correctly constructed and presented. It could only be enriched with information on how many collected surveys were filled in in 100 percent. The results of the research were presented legibly and without unnecessary repetitions. The discussion refers to the results obtained. It was written in a balanced way, the authors try to relate to the results obtained, they take into account the cultural and organizational specificity of health care systems. The conclusions could be presented in the form of points. References correctly selected.
Author Response
Point 1: I read with pleasure and great interest the work devoted to the extremely important issue which is quality of care. It analyzes in an innovative way the phenomenon taking into consideration non-clinical conditions. Moreover, unlike most studies in the field of quality of care that focus on recipients of services, it focuses on the persons responsible for arranging services, i.e. the management.
Sufficient introduction, the authors could, however, provide accurate information in which countries the tool was used. Work methodology was correctly constructed and presented.
Response 1: The introduction was enriched adding the countries in which the HLHO-10 had been already used.
Point 2: It could only be enriched with information on how many collected surveys were filled in in 100 percent.
Response 2: The requested data has been added
Point 3: The results of the research were presented legibly and without unnecessary repetitions. The discussion refers to the results obtained. It was written in a balanced way, the authors try to relate to the results obtained, they take into account the cultural and organizational specificity of health care systems.
The conclusions could be presented in the form of points.
Response 3: In our opinion, the conclusions presented in the form of points could limit the legibility, but we will change this section, if this should be deemed essential.
Round 2
Reviewer 1 Report
I appreciate the additional analysis and revisions the authors made to respond to my earlier comments. However, my earlier concern about the fundamental design of the study remains. My interpretation of the findings is that that rather than showing that HLHO is associated with quality, the researchers have shown that middle managers who think their hospitals are health literate also think their units delivery adequate non-clinical quality of care. The limitation that the authors added (lines 365-368) does not adequately cover this limitation. They focused on the fact that the data were collected at the same time. The real problem is that both variables are subjective assessments of the same individuals, rather than using an objective measurement - or at least someone else's subjective assessment - of the quality of care.
In their response the authors described how they minimized social desirability bias by selecting middle managers as their respondents. I think the limitation of social desirability and its mitigation by using middle managers should be added to the manuscript.
Here are some additional suggestions on the revised manuscript.
- The term “years in charge,” which has been now translated as “years in chief” in the new Table 5, is still problematic. At least for American offices, “years in department” or “years in unit” would be more understandable.
- The English of the newly inserted text (e.g., 319-324) needs improving.
- In response to my comments, the authors added four paragraphs reconciling the results of this study with the Turkish and German studies. While very interesting, I believe this is excessive and recommend reducing this substantially.
Author Response
Point 1: I appreciate the additional analysis and revisions the authors made to respond to my earlier comments. However, my earlier concern about the fundamental design of the study remains. My interpretation of the findings is that that rather than showing that HLHO is associated with quality, the researchers have shown that middle managers who think their hospitals are health literate also think their units delivery adequate non-clinical quality of care. The limitation that the authors added (lines 365-368) does not adequately cover this limitation. They focused on the fact that the data were collected at the same time. The real problem is that both variables are subjective assessments of the same individuals, rather than using an objective measurement - or at least someone else's subjective assessment - of the quality of care.
Response 1: Thanks for the comment. We are aware that the study design has limitations related to the use of subjective measures of OHL and non-clinical quality of care that have been administered to the same sample of middle managers. We have explored the availability of data regarding non-clinical quality of care measured using objective measurement tools or collected in surveys that included other target groups. Unfortunately, we have not found adequate data about the hospitals included in our study (private hospitals are generally not involved in these kind of surveys), about the items that we considered (survey generally address different aspects) or microdata. Moreover, in some cases, data were too old to be compared with those of our study. For these reasons, at the moment we are not able to add data or other analysis to address the concern expressed by the referee. For these reasons, we have added this aspect in the limitation of the study (lines 388-391) and in the conclusion (line 493).
Point 2 In their response the authors described how they minimized social desirability bias by selecting middle managers as their respondents. I think the limitation of social desirability and its mitigation by using middle managers should be added to the manuscript.
Response 2 This aspect has been added in the limitation
Point 3 The term “years in charge,” which has been now translated as “years in chief” in the new Table 5, is still problematic. At least for American offices, “years in department” or “years in unit” would be more understandable.
Response 3: the term has been changed as suggested.
Point 4: The English of the newly inserted text (e.g., 319-324) needs improving.
Response 4: The English of the newly inserted text has been improved.
Point 5: In response to my comments, the authors added four paragraphs reconciling the results of this study with the Turkish and German studies. While very interesting, I believe this is excessive and recommend reducing this substantially.
Response 5: This part has been reduced in length.
Reviewer 3 Report
Congratulations to the authors for their work.
Best regards
Author Response
Response to Reviewer 3 Comments
Point 1: Congratulations to the authors for their work.
Best regards
Response 1: Thank you very much for your kind cooperation and for your suggestions